# Maternal selenium deficiency was positively associated with the risk of selenium deficiency in children aged 6–59 months in rural Zimbabwe

Beaula Mutonhodza[1]*, Muneta G. Manzeke-Kangara[2], Elizabeth H. Bailey[3]*, Tonderayi M. Matsungo[1©], Prosper Chopera[1©]

1 Department of Nutrition, Dietetics and Food Sciences, University of Zimbabwe, Harare, Zimbabwe, 2 Rothamsted Research, West Common, Harpenden, United Kingdom, 3 School of Biosciences, Sutton Bonington Campus, University of Nottingham, Loughborough, Leicestershire, United Kingdom

© These authors contributed equally to this work.
* bmutonhodza@science.uz.ac.zw (BM); liz.bailey@nottingham.ac.uk (EHB)

## Abstract

There is growing evidence showing the existence of selenium (Se) deficiency among women and children in sub-Saharan Africa. Unfortunately, the key drivers of Se deficiency are not clearly understood. This study assessed the determinants of Se deficiency among children aged 6–59 months and Women of Reproductive Age (WRA), in Zimbabwe. This cross-sectional biomarker study was conducted in selected districts in rural Zimbabwe (Murewa, Shamva, and Mutasa). Children aged 6–59 months (n = 683) and WRA (n = 683), were selected using a systematic random sampling approach. Venous blood samples were collected, processed, and stored according to World Health Organization (WHO) guidelines. Plasma selenium concentration was measured using inductively coupled plasma-mass spectrometry (ICP-MS). Anthropometric indices were assessed and classified based on WHO standards. Demographic characteristics were adapted from the Zimbabwe Demographic Health Survey standard questionnaire. Multiple logistic regression analysis showed that children whose mothers were Se deficient were 4 times more likely to be Se deficient compared to those whose mothers were Se adequate (OR = 4.25; 95% CI; 1.55–11.67; p = 0.005). Girl children were 3 times more likely to be Se deficient compared to boys (OR = 2.84; 95% CI; 1.08–7.51; p = 0.035). Women producing maize for consumption were 0.5 times more likely to be Se deficient than non-producers (OR = 0.47; 95% CI; 0.25–0.90; p = 0.022). The risk of Se depletion in children was amplified by maternal deficiency. Therefore, initiation of maternal multiple micronutrient supplementation from preconception through lactation is beneficial to both children and women.

## 1. Introduction

Selenium deficiency is widespread among children and women in sub-Saharan Africa [1, 2] and has been implicated as a potential causal factor of growth faltering in children [3, 4], and

**Data Availability Statement:** All raw data required to replicate the results of the study has been

provided as part of the submitted article, in a format that can be accessed without restrictions.

**Funding:** Authors acknowledge funding from the UK Research and Innovation (UKRI) Global Challenges Research Fund (GCRF) [grant number EP/T015667/1; "Translating GeoNutrition: Reducing mineral micronutrient deficiencies (MMNDs) in Zimbabwe". The work was also supported in part by Bill & Melinda Gates Foundation grant INV-009129 through the GeoNutrition project. Under the grant conditions of the Foundation, a Creative Commons Attribution 4.0 Generic License has already been assigned to the Author's Accepted Manuscript version that might arise from this submission. The funders had no role in study design, data collection and analysis, decision to publish, or preparation of the manuscript.

**Competing interests:** The authors have declared that no competing interests exist.

fertility impairments in women [5]. Worldwide, up to one in seven people are estimated to have low dietary Se intake [6], and c. 0.5 to 1 billion people are Se deficient [7, 8]. Estimated low dietary mineral supply of Se in Africa [9] may have considerable regional public health significance [10]. The greater prevalence of Se deficiency highlighted in sub-Saharan African countries [1], can be exacerbated by the human immunodeficiency virus [11] which is prevalent in many African settings [12], posing further potential public health concerns. Viral infection simultaneously increases the demand for micronutrients and causes their loss, exacerbating deficiency [11].

Diet is the main source of Se [13] with meat and meat products rich in Se [14]. However, across sub-Saharan Africa diets are predominantly crop-based and it is these sources that provide the majority of dietary Se [15]. Diets in sub-Saharan Africa consist primarily of carbohydrates [16]. In many parts of Africa, rural diets are frequently monotonous, consisting mainly of starchy foods such as grains, tubers, and roots but with limited or negligible intake of animal-source foods [17]. The consumption of animal foods such as meat, poultry, and fish is limited, mainly because of economic, cultural, and religious constraints [15]. Plants can be classified into three main groups based on the Se concentrations in their tissues; non-accumulators, accumulators, and hyperaccumulators [18]. Non-accumulating plants such as grains and grasses contain lower concentrations of Se [18, 19]. Maize grain in sub-Saharan African countries has a sub-optimal Se concentration of < 50 μg/kg dry mass [2] and is not likely to meet human requirements [20]. Typically, Se deficiency is a consequence of inadequate dietary Se intake [14, 21, 22], however, there are multiple proximal risk factors: inflammation [23, 24], body mass index (BMI) [25, 26], gender, age, protein malnutrition [27] and dietary diversity [28]. Socioeconomic and environmental distal factors, such as wealth status and rural or urban residence, influence Se status [22, 29]. In low and middle-income countries macroeconomic volatility is common and severe negative economic shocks can substantially increase poverty, food insecurity, and risks of inadequate dietary diversity [30]. Preliminary estimates for Zimbabwe suggested that the number of extremely poor reached 7.9 million in 2020, 49% of the population [31]. Government data for Zimbabwe [32] indicates that 4% of children between 6–23 months of age receive a minimum acceptable diet and 16% consume the minimum number of food groups recommended for their age. It also reports that the proportion of women of reproductive age (WRA) consuming at least four food groups was 44% [32], and more than 50% of the population is affected by micronutrient deficiencies (MNDs) [33].

Human Se deficiency has been reported previously in Zimbabwe [34, 35]. However, the key drivers of Se deficiency are not clearly understood. Data for this study was collected in rural Zimbabwe as part of a baseline study for a micronutrient biomarker survey. The work was guided by the UNICEF's Conceptual Framework on the Determinants of Maternal and Child Nutrition, 2020 [36] and explored the immediate, underlying, basic, and enabling causes of Se deficiency among children aged 6–59 months and WRA in selected districts (Murewa, Shamva, and Mutasa).

## 2. Materials and methods

### 2.1. Ethical statements

The study was conducted in line with the Declaration of Helsinki. Ethical approval was obtained from the Institutional Review Boards of the University of Nottingham (Reference#446–1912) and the Medical Research Council of Zimbabwe (MRCZ/A/2575). Shipping permissions, including a material transfer agreement were secured. Research approval was awarded by local government officials and the Ministry of Health at the provincial, district,

clinic, and village levels. Written informed consent and assent were obtained from all WRA and all child participants before the commencement of data collection, respectively.

## 2.2. Sampling

A detailed description of the methods has been reported elsewhere [35, 37, 38]. In summary, the current paper presents data from a cross-sectional study on the determinants of Se deficiency in children aged 6–59 months (n = 683) and in WRA (n = 683) from three rural districts; Murewa (17.6502˚S, 31.7787˚E), Shamva (17.04409˚S, 31.6739˚E), and Mutasa (18.6155˚S, 32.6730˚E) in Zimbabwe. Data collection was between 25 October 2021 and 30 January 2022. The sampling design was nested at the level of the National Demographic Health Survey (DHS) sampling approach [33]. Thirty Enumeration areas (EAs) proportional to the most recently recorded population [39], were selected per district. Random systematic sampling without replacement was used to select 10 eligible households from each EA; the Kish Grid [40] was used for the within-household selection of multiple eligible individuals. Participants were directed to the nearest health facility for data and sample collection by trained personnel.

## 2.3. Data and sample management

A temporary laboratory was established at each collection site to minimize contamination, facilitate accurate record keeping, and for traceability of samples. Strict quality control measures were followed as guided by the CDC [41]. Each participant was assigned a unique numeric identity that was used on data capture forms, sample collection materials, and subsequent analyses to maintain anonymity. Passcode-protected tablets with Kobo Toolbox software (Android v2022.1.2) were used to capture demographic and specimen data.

## 2.4. Data collection and analysis

**2.4.1. Demographic characteristics.** A questionnaire adapted from ZDHS [33] was used to collect household demographic data. The questionnaire also assessed socioeconomic characteristics (education level, marital status, and income status), health status, agricultural, water sanitation and hygiene (WASH), and infant and young child feeding practices as adapted from the UNICEF conceptual framework (Fig 1).

**2.4.2. Anthropometry.** Weight, recumbent length, and height were measured according to World Health Organization (WHO) standard protocols [42] and standardized as required for nutrition assessments [43, 44]. The anthropometric indices, namely height-for-age Z-score (HAZ), weight-for-height Z-score (WHZ), and weight-for-age Z-score (WAZ), for children, were generated using the Emergency Nutrition Assessment software for SMART 2011 [45]. Wasting was defined as WHZ below -2 Standard Deviations (SD), stunting as HAZ below-2SD, underweight as WAZ below -2SD, and overweight as WHZ above +2SD [44]. Birth weight was obtained from the infant's health cards, birth weight below 2.5 kg was defined as low birth weight (LBW) [46]. Body mass index (BMI) was calculated and classified for WRA according to WHO guidelines; BMI below 18.5 was considered as underweight; 18.5–24.9, normal weight; 25.0–29.9, overweight; 30.0–34.9, obesity class I; 35.0–39.9, obesity class II; and above 40 defined as morbid obesity [47]. Maternal short stature was defined as height below 145 cm [48]. While the reference reproductive age for optimal birth outcomes was 18–34 years [49].

**2.4.3. Blood.** A venous blood sample (6 mL blood) was collected from children 6–59 months and WRA according to the WHO blood collection guidelines [50]. Blood was centrifuged to isolate plasma in the field at 3000 rpm for 10 minutes. Center for Disease Control &

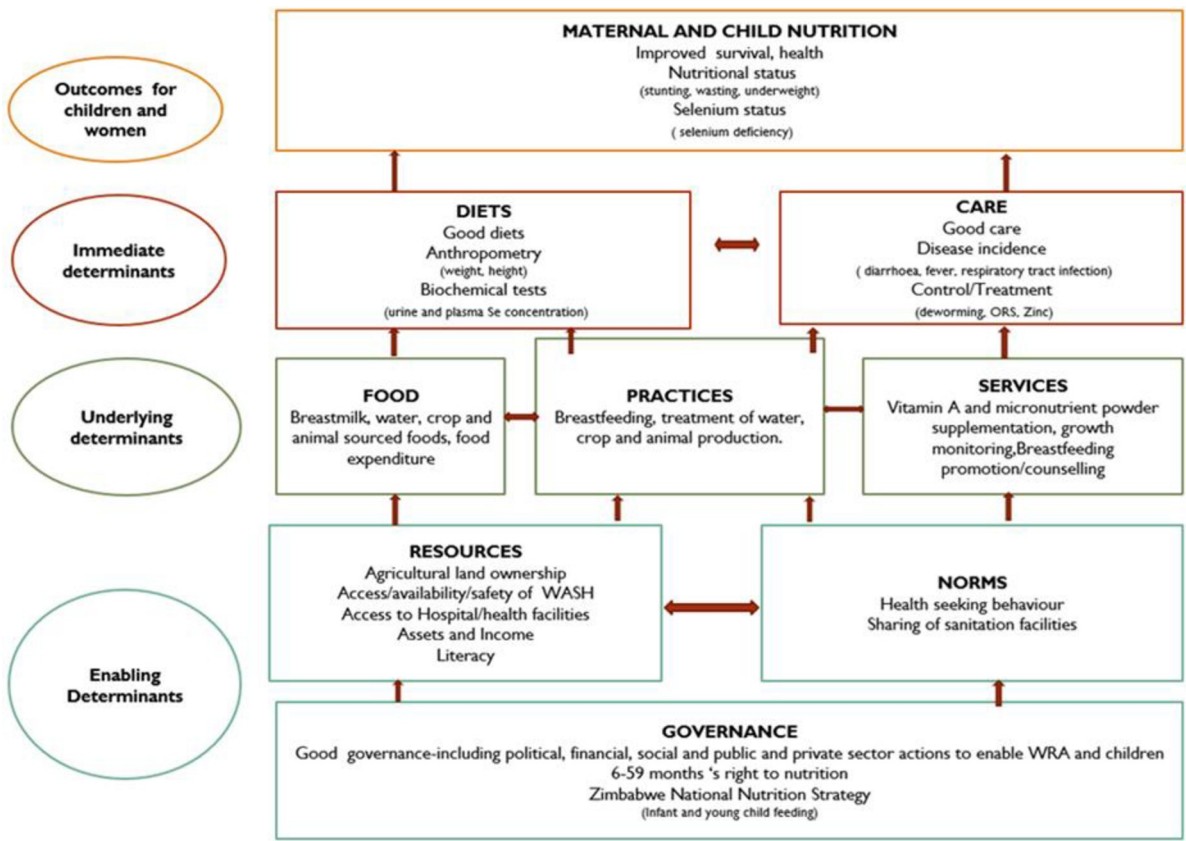

**Fig 1. Theoretical framework adapted from UNICEF conceptual framework on the Determinants of Maternal and Child Nutrition 2020 [36].**

Prevention (CDC) guidelines were followed to reduce the risk of hemolysis [51]. A cold chain was maintained during sample transportation and the plasma was stored at -80˚C. Biomarkers of inflammation; C-reactive protein and alpha-1-acid glycoprotein were analyzed by a sandwich ELISA as adapted from Erhardt *et al.*, (2004) [52]. Quantification of plasma Se concentration was conducted using inductively coupled plasma-mass spectrometry (ICP-MS) as described by Belay *et al.* (2020) [53] and Phiri *et al.* (2019) [22]. The limit of detection (LOD) and the limit of quantification (LOQ) were 0.029 and 0.096 µg/L, respectively. All the sample observations were above the LOD/LOQ limits. Average Se recovery was 99% and 102% for reference materials (Seronorm L-1 (Lot 1801802) and Seronorm L-2 (Lot 1801803); Nycomed Pharma AS, Billingstad, Norway), respectively. No significant correlation was observed between plasma Se concentration and any of the inflammation biomarkers measured (S1 Table), thus no correction was applied for inflammation [50, 54].

## 2.5. Data analysis

Selenium deficiency was determined against plasma Se concentration thresholds of 70 µg/L [53] for children aged 6–59 months and women aged 15–49 years. Statistical analysis used SPSS for Windows version 20 (IBM, New York, USA). Exploratory data analysis was done using, quantile–quantile (QQ) plots to check for outliers and data normality [55]. Selenium concentration data was collapsed into categorical data (Se adequate and Se deficient) and was

considered as the dependent variable singly for children and WRA. Association between the dependent variable and socio-demographic factors (age, sex, education level, marital status, agricultural and WASH factors) and anthropometry indicators were evaluated using the Pearson Chi-square test and two-sided p-values < 0.05 were considered statistically significant. Significant variables (determinants) from the Pearson Chi-square test were further analyzed using logistic regression, with the removal criterion (pR) value set at 0.10 and 0.05 as the entry criterion ($p_E$) value by the enter method to establish predictors of Se deficiency. All statistical measures were at 95% confidence interval.

## 3. Results

### 3.1. Demographic and health characteristics of participants

A total of 683 mother-child pair Se concentration measurements (dependent variable) were matched and analyzed. The sample size in the three districts was proportionate across the two demographic groups as was the boy/girl ratio of the children. The median (Q1, Q3) age for the children was 29 months [18, 44], and that for women was 30 years [24, 37]. The majority of WRA who participated in the study were married (88.7%) and had acquired secondary-level education qualifications (66.2%). A few households (8.5%) earned a monthly household income adequate to meet the total consumption poverty line, set at 63.50 United States dollars per person, as of August 2021 [56]. Land ownership was ≥ 5 hectares for most of the households (60.8%). Maize *(Zea mays)* was the most predominant (64%) crop grown for household consumption. Most households had access to water, sanitation, and hygiene (WASH) facilities, with 78.4% having adequate water in the 30 days preceding the survey date, despite the water source being located off-premises for most of the households (66.5%). Almost all households (96.9%) had access to toilet facilities.

Close to a third (27.7%) of the children were classified as stunted, 2.8% were wasted and 13.5% were underweight. The rate of LBW was 9.0%. A high prevalence of breastfeeding was observed in the sampled population, with 69.4% of children exclusively breastfed up to 6 months as recommended by WHO. Vitamin A supplementation was high at >70%, while multiple micronutrient powder (MNP) supplementation coverage was low (9.4%) in children. Disease prevalence 2 weeks preceding the survey date indicated diarrhea had the lowest prevalence (24.6%), followed by fever (34.2%), with respiratory infection having the highest prevalence (35.5%). The prevalence of anemia (Hb <11 g/dL) in children was high at 28.1%, and 96.2% of the children were Se deficient. In WRA, the prevalence of anemia (Hb <12 g/dL) was 19.8% and that of Se deficiency was 70.1%. Based on BMI, a few women (10.9%) were underweight, the majority (52.1%) had normal weight and the proportions of overweight, class I, class II, and morbid obesity were 26.6%, 7.1%, 1.9%, and 1.3%, respectively.

### 3.2. Sociodemographic factors and Se deficiency

**3.2.1. Children 6–59 months.**   Residency in Murewa district (p <0.001), being a girl (p = 0.008), the use of unimproved dug wells as a source of drinking water (p = 0.014), and the production of maize (p <0.001), cowpeas *(Vigna unguiculata)* (p <0.001), groundnuts *(Arachis hypogaea)* (p <0.001), sweet potatoes *(Ipomoea batatas)* (p <0.001), sugar beans *(Phaseolus vulgaris)* (p = 0.005) and onions *(Allium cepa)* (p = 0.002), for household consumption (Table 1) were significantly associated with Se deficiency in children (Pearson's $\chi^2$ test).

**3.2.2. Women of reproductive age.**   Residency in Murewa district (p <0.001), reproductive age of 18–34 years (p = 0.036), monthly income of below 10 USD (p <0.001), consumption of water from unimproved dug wells (p <0.001) and production of maize (p <0.001), production of cowpeas (p <0.001), production of groundnuts (p <0.001), production of sweet

**Table 1. Sociodemographic characteristics of children 6–59 months in rural Zimbabwe by Se status.**

| Variable | Total n (%) | ‡Se-adequate n (%) | §Se deficient n (%) | P-value† |
|---|---|---|---|---|
| **District** | | | | |
| **Mutasa** | 203 (29.7) | 17 (8.4) | 186 (91.6) | <0.001* |
| **Shamva** | 257 (37.6) | 5 (1.9) | 252 (98.1) | |
| **Murewa** | 223 (32.7) | 4 (1.8) | 219 (98.2) | |
| **Sex** | | | | |
| **Boy** | 332 (49.7) | 19 (5.7) | 313 (94.3) | 0.008* |
| **Girl** | 336 (50.3) | 6 (1.8) | 330 (98.2) | |
| **Age group (months)** | | | | |
| **6–8** | 25 (3.8) | 1 (4.0) | 24 (96.0) | 0.915 |
| **9–11** | 32 (4.8) | 0 (0) | 32 (100) | |
| **12–17** | 91 (13.7) | 4 (4.4) | 87 (95.6) | |
| **18–23** | 102 (15.4) | 5 (4.9) | 97 (95.1) | |
| **24–35** | 156 (23.5) | 5 (3.2) | 151 (96.8) | |
| **36–47** | 132 (19.9) | 6 (4.5) | 126 (95.5) | |
| **48–59** | 125 (18.9) | 4 (3.2) | 121 (96.8) | |
| **Number of children under 5 years in the household** | | | | |
| **1** | 489 (71.6) | 23 (4.7) | 466 (95.3) | 0.073 |
| **>1** | 194 (28.4) | 3 (1.5) | 191 (98.5) | |
| **Household size** | | | | |
| **≤4** | 265 (38.8) | 7 (2.6) | 258 (97.4) | 0.226 |
| **>4** | 418 (61.2) | 19 (4.5) | 399 (95.5) | |
| **Household monthly income (USD)** | | | | |
| **<10** | 58 (8.5) | 2 (3.4) | 56 (96.6) | 0.328 |
| **10–50** | 312 (45.7) | 7 (2.2) | 305 (97.8) | |
| **51–110** | 170 (24.9) | 10 (5.9) | 160 (94.1) | |
| **120–210** | 85 (12.4) | 4 (4.7) | 81 (95.3) | |
| **>220** | 58 (8.5) | 3 (5.2) | 55 (94.8) | |
| **Agricultural land ownership** | | | | |
| **No** | 268 (39.2) | 9 (3.4) | 259 (96.6) | 0.687 |
| **Yes** | 415 (60.8) | 17 (4.1) | 398 (95.9) | |
| **Livestock ownership** | | | | |
| **Chicken /poultry** | | | | |
| **1** | 139 (20.4.) | 7 (5.0) | 132 (95.0) | 0.454 |
| **>1** | 544 (79.6) | 19 (3.5) | 525 (96.5) | |
| **Common crops grown for consumption** | | | | |
| *Maize* | | | | |
| **No** | 233 (36.0) | 17 (7.3) | 216 (92.7) | <0.001* |
| **Yes** | 414 (64.0) | 6 (1.4) | 408 (98.6) | |
| *Cowpeas* | | | | |
| **No** | 337 (53.8) | 21 (6.2) | 316 (93.8) | <0.001* |
| **Yes** | 289 (46.2) | 2 (0.7) | 287 (99.3) | |
| *Groundnuts* | | | | |
| **No** | 321 (51.3) | 20 (6.2) | 301 (93.8) | <0.001* |
| **Yes** | 305 (48.7) | 3 (1.0) | 302 (99.0) | |
| *Sugar beans* | | | | |
| **No** | 425 (67.9) | 22 (5.2) | 403 (94.8) | 0.005* |

(Continued)

**Table 1.** (Continued)

| Variable | Total n (%) | ‡Se-adequate n (%) | §Se deficient n (%) | P-value† |
|---|---|---|---|---|
| Yes | 201 (32.1) | 1 (0.5) | 200 (99.5) | |
| *Sweet potatoes* | | | | |
| No | 345 (55.1) | 21 (6.1) | 324 (93.6) | <0.001* |
| Yes | 281 (44.9) | 2 (0.7) | 279 (99.3) | |
| *Onions* | | | | |
| No | 417 (66.6) | 22 (5.3) | 395 (94.7) | 0.002* |
| Yes | 209 (33.4) | 1 (0.5) | 208 (99.5) | |
| **Unimproved dug wells as a source of drinking water** | | | | |
| No | 485 (71.0) | 24 (4.9) | 461 (95.1) | 0.014* |
| Yes | 198 (29.0) | 2 (1.0) | 196 (99.0) | |
| **Location of water source** | | | | |
| **Off-premise (elsewhere)** | 454 (66.5) | 16 (3.5) | 438 (96.5) | 0.443 |
| **In-house (own dwelling)** | 24 (3.5) | 0 (0) | 24 (100) | |
| **On-premise (own yard/plot)** | 205 (30.0) | 10 (4.9) | 195 (95.1) | |
| **Insufficient water in the past month** | | | | |
| No | 536 (78.4) | 19 (3.6) | 517 (96.4) | 0.641 |
| Yes | 147 (21.6) | 7 (4.8) | 140 (95.2) | |
| **Treatment of drinking water** | | | | |
| No | 604 (88.4) | 25 (4.1) | 579 (95.9) | 0.243 |
| Yes | 79 (11.6) | 1 (1.3) | 78 (11.9) | |
| **Toilet facility** | | | | |
| No | 21 (3.1) | 1 (4.8) | 20 (95.2) | >0.999 |
| Yes | 662 (96.9) | 25 (3.8) | 637 (96.2) | |
| **Toilet facility shared with other households** | | | | |
| No | 489 (71.6) | 19 (3.9) | 470 (96.1) | >0.999 |
| Yes | 194 (28.4) | 7 (3.6) | 187 (96.4) | |
| **Toilet facility on-premise** | | | | |
| No | 662 (96.9) | 25 (3.8) | 637 (96.2) | >0.999 |
| Yes | 21 (3.1) | 1 (4.8) | 20 (95.2) | |
| **Overall, Se deficiency prevalence:** | 683 (100) | 26 (3.8) | 657 (96.2) | |

Notes:

§ Plasma Se concentration <70 μg/L;

‡ Plasma Se concentration ≥70 μg/L;

*Significant at P <0·05;

†P value from Pearson's χ2 test.

Age categories and average household size used are based on previous demographic health surveys (33)

potatoes (p <0.001), production of sugar beans (p <0.001) and production of onions (p <0.001) for household consumption, (Table 2) were significantly associated with Se deficiency in WRA (Pearson's χ2 test).

## 3.3. Factors associated with Se deficiency

**3.3.1. Children 6–59 months.** Child Se status was significantly associated with maternal Se status (p <0.001). The proportion of Se-deficient children was higher (98.7%) in Se-

**Table 2. Sociodemographic characteristics of WRA in rural Zimbabwe by Se status.**

| Variable | Total n (% of category) | ‡Se adequate n (%) | §Se deficient n (%) | P-value† |
|---|---|---|---|---|
| **District** | | | | |
| **Mutasa** | 203 (29.7) | 108 (53.2) | 95 (46.8) | <0.001* |
| **Shamva** | 257 (37.6) | 62 (24.1) | 195 (75.9) | |
| **Murewa** | 223 (32.7) | 34 (15.2) | 189 (84.8) | |
| **Reproductive age (years)** | | | | |
| **<18** | 15 (2.2) | 9 (60.0) | 6 (40.0) | |
| **18–34** | 426 (62.4) | 123 (28.9)) | 303 (71.1) | 0.036* |
| **≥35** | 242 (35.4) | 72 (29.8) | 170 (70.2) | |
| **Marital status** | | | | |
| **Married monogamy** | 567 (83.0) | 170 (30.0) | 397 (70.0) | 0.888 |
| **Married polygamy** | 39 (5.7) | 13 (33.3) | 26 (66.7) | |
| **Separated/divorced** | 46 (6.7) | 14 (30.4) | 32 (69.6) | |
| **Single /never married** | 7 (1.0) | 2 (28.6) | 5 (1.0) | |
| **Widowed** | 24 (3.5) | 5 (20.8) | 19 (79.2) | |
| **Education status** | | | | |
| **Tertiary** | 7 (1.0) | 4 (57.1) | 3 (42.9) | 0.274 |
| **Advanced level** | 6 (0.9) | 3 (50.0) | 3 (50.0) | |
| **Ordinary level** | 452 (66.2) | 136 (30.1) | 316 (69.9) | |
| **Primary** | 205 (30.0) | 59 (28.8) | 146 (71.2) | |
| **No formal education** | 13 (1.9) | 2 (15.4) | 11 (84.6) | |
| **Number of children under 5 years in the household** | | | | |
| **1** | 489 (71.6) | 155 (31.7) | 334 (68.3) | 0.115 |
| **>1** | 194 (28.4) | 49 (25.3) | 145 (74.7) | |
| **Household size** | | | | |
| **≤4** | 265 (38.8) | 76 (28.7) | 189 (71.3) | 0.608 |
| **>4** | 418 (61.2) | 128 (30.6) | 290 (69.4) | |
| **Household monthly income (USD)** | | | | |
| **<10** | 58 (8.5) | 9 (15.5) | 49 (84.5) | <0.001* |
| **10–50** | 312 (45.7) | 75 (24.0) | 237 (76.0)) | |
| **51–110** | 170 (24.9) | 67 (39.4) | 103 (60.6) | |
| **120–210** | 85 (12.4) | 34 (40.0) | 51 (60.0) | |
| **>220** | 58 (8.5) | 19 (32.8) | 39 (67.2) | |
| **Agricultural land ownership** | | | | |
| **No** | 268 (39.2) | 78 (29.1) | 190 (70.9) | 0.726 |
| **Yes** | 415 (60.8) | 126 (30.4) | 289 (69.6) | |
| **Common crops grown for consumption** | | | | |
| *Maize* | | | | |
| **No** | 233 (36.0) | 17 (7.3) | 216 (92.7) | <0.001* |
| **Yes** | 414 (64.0) | 6 (1.4) | 408 (98.6) | |
| *Cowpeas* | | | | |
| **No** | 337 (53.8) | 21 (6.2) | 316 (93.8) | <0.001* |
| **Yes** | 289 (46.2) | 2 (0.7) | 287 (99.3) | |
| *Groundnuts* | | | | |
| **No** | 321 (51.3) | 20 (6.2) | 301 (93.8) | <0.001* |
| **Yes** | 305 (48.7) | 3 (1.0) | 302 (99.0) | |
| *Sugar beans* | | | | |

*(Continued)*

**Table 2.** (Continued)

| Variable | Total n (% of category) | ‡Se adequate n (%) | §Se deficient n (%) | P-value† |
|---|---|---|---|---|
| No | 425 (67.9) | 22 (5.2) | 403 (94.8) | <0.001* |
| Yes | 201 (32.1) | 1 (0.5) | 200 (99.5) | |
| *Sweet potatoes* | | | | |
| No | 345 (55.1) | 21 (6.1) | 324 (93.6) | <0.001* |
| Yes | 281 (44.9) | 2 (0.7) | 279 (99.3) | |
| *Onions* | | | | |
| No | 417 (66.6) | 22 (5.3) | 395 (94.7) | <0.001* |
| Yes | 209 (33.4) | 1 (0.5) | 208 (99.5) | |
| **Livestock commonly reared** | | | | |
| **Chicken /poultry** | | | | |
| 1 | 139 (20.4) | 47 (33.8) | 92 (66.2) | 0.299 |
| >1 | 544 (79.6) | 157 (28.9) | 387 (71.1) | |
| **Unimproved dug wells as a source of drinking water** | | | | |
| No | 485 (71.0) | 168 (34.6) | 317 (65.4) | <0.001* |
| Yes | 198 (29.0) | 36 (18.2) | 162 (81.8) | |
| **Location of water source** | | | | |
| Off-premise | 454 (66.5) | 137 (30.2) | 317 (69.8) | 0.658 |
| In-house | 24 (3.5) | 9 (37.5) | 15 (62.5) | |
| On-premise | 205 (30.0) | 58 (28.3) | 147 (71.7) | |
| **Insufficient water in the past month** | | | | |
| No | 536 (78.4) | 154 (28.6) | 382 (71.4) | 0.103 |
| Yes | 147 (21.6) | 50 (34.0) | 97 (66.0) | |
| **Treatment of drinking water** | | | | |
| No | 604 (88.4) | 184 (30.5) | 420 (69.5) | 0.264 |
| Yes | 79 (11.6) | 20 (25.3) | 59 (74.7) | |
| **Toilet facility** | | | | |
| No | 21 (3.1) | 3 (14.3) | 18 (85.7) | 0.147 |
| Yes | 662 (96.9) | 201 (30.4) | 461 (69.6) | |
| **Toilet facility shared with other households** | | | | |
| No | 489 (71.6) | 140 (28.6) | 349 (71.4) | 0.267 |
| Yes | 194 (28.4) | 64 (33.0) | 130 (67.0) | |
| **Toilet facility on-premise** | | | | |
| No | 662 (96.9) | 195 (29.5) | 467 (70.5) | 0.225 |
| Yes | 21 (3.1) | 9 (42.9) | 12 (57.1) | |
| **Prevalence of Se deficiency (WRA):** | 683 (100) | 204 (29.9) | 479 (70.1) | |

Notes:

§ Plasma Se concentration <70 μg/L;

‡ Plasma Se concentration ≥70 μg/L;

*Significance level P <0·05;

†P value from Pearson's Chi square test.

Age categories and average household size used are based on previous demographic health surveys [33]. Reproductive age represents the stage of conception where 18–34 years is the reference for optimal reproductive function established from the NHANES (2011–2012)sourced from [57].

deficient mothers compared to Se-adequate mothers (90.2%) (Table 3), with older children being significantly more Se-deficient compared to the younger children. Among the Se-deficient mothers, children aged 6–8 months were the least Se-deficient (94.4%, p >0.999) while those in the age range 24–35 months were the most Se-deficient (100%, p = 0.004) S2 Table.

**3.3.2. Women of reproductive age.** There were no significant correlations between Se deficiency and height, body mass index, and anaemia in WRA (Table 4).

### 3.4. Predictors of Se deficiency in children 6–59 months and WRA

Being a girl (p = 0.035) and having a Se-deficient mother (p = 0.005) were predictors of Se deficiency in children (Table 5). Children whose mothers were Se deficient were 4 times more likely to be Se deficient compared to those whose mothers were not Se deficient (OR = 4.25; 95% CI; 1.55–11.67; p = 0.005) and female children were 3 times more likely to be Se deficient compared to male children (OR = 2.84; 95% CI; 1.08–7.51; p = 0.035). Women producing mainly maize for consumption were 0.5 times more likely to be Se deficient than those not growing maize mainly for consumption (OR = 0.47; 95% CI; 0.25–0.90; p = 0.022).

## 4. Discussion

### 4.1. Overview of the current study findings

The study sought to determine the predictors of Se deficiency in children aged 6–59 months and in WRA. It was evident that Se status was inversely associated with immediate determinants such as maternal Se deficiency while underlying determinants included food production practices and WASH, exacerbated by enabling determinants such as residency, income status, and gender. Determinants of inadequate Se status in both women and children included residency in the Murewa district, the use of unimproved dug wells as sources of drinking water, and the production of maize, sugar beans, groundnuts, cowpeas, sweet potatoes, and onions for household consumption. Maternal Se deficiency and being a girl were positively associated with Se deficiency in children. Reproductive age (18–34 years) and low monthly household income were positively associated with Se deficiency in WRA. Predictors of plasma Se status in children were maternal Se status and being a girl, whereas, in women, it was the production of maize as the main crop for consumption (Fig 2).

### 4.2. Sex differences in Se deficiency

Our results show that girls were more likely to be Se deficient than boys, consistent with previous studies in Vietnam [58] and Zimbabwe [34]. In contrast, studies in Ethiopia found no sex-related differences in Se status [3]. The micronutrient survey conducted in Zimbabwe also indicated a slightly higher prevalence of MNDs in girls than boys [59]. Exploration of sex-based factors that influence Se intake is outside the scope of this study however, the disparity in the prevalence of Se deficiency in girls and boys could be explained by physiological differences in the expression of deficiencies [60]; sexual dimorphic regulation of Se metabolism and selenoprotein expression, namely the trans-selenation pathway by sex hormones strongly implies that selenomethionine metabolism and its consequent selenocysteine formation and availability for selenoprotein synthesis are not the same in both sexes [61]. Selenium is concentrated in male gonads which could explain why male children had lower deficiency than female children [62]; or gender-based vulnerabilities [63, 64] influencing food intake [64]. In India, girls were more likely to be neglected than boys to receive nutritious diets [65], female children were breastfed for a shorter duration and had lower consumption of dairy food compared to male children [66]. These differences can be attributed to gender incongruence in the intra-

**Table 3. Nutritional status and morbidities in children aged 6–59 months in rural Zimbabwe by Se status.**

| Variable | Total n (% of category) | ‡ Se adequate n (%) | §Se deficient n (%) | P-value[†] |
|---|---|---|---|---|
| **Stunted (HAZ)** | | | | |
| Below -2SD | 177 (27.7) | 3 (1.7) | 174 (98.3) | 0.106 |
| -2SD and above | 462 (72.3) | 21 (4.5) | 441 (95.5) | |
| **Wasting (WHZ)** | | | | |
| Below -2SD | 18 (2.8) | 0 (0) | 18 (100) | 0.641 |
| -2SD and above | 623 (97.2) | 24 (3.9) | 599 (96.1) | |
| **Underweight (WAZ)** | | | | |
| Below -2SD | 89 (13.5) | 4 (4.5) | 85 (95.5) | >0.999 |
| -2SD and above | 572 (86.5) | 22 (3.8) | 550 (96.2) | |
| **Low Birth Weight (kg)** | | | | |
| <2.5 | 58 (9.0) | 1 (1.7) | 57 (98.3) | 0.498 |
| ≥2.5 | 590 (91.0) | 24 (4.1) | 566 (95.9) | |
| **Exclusive Breastfeeding** | | | | |
| No | 204 (30.6) | 4 (2.0) | 200 (98.0) | 0.124 |
| Yes | 462 (69.4) | 21 (4.5) | 441 (95.5) | |
| **Child still breastfeeding** | | | | |
| No | 507 (76.0) | 19 (3.7) | 488 (96.3) | >0.999 |
| Yes | 160 (24.0) | 6 (3.8) | 154 (96.2) | |
| **Vitamin A supplementation** | | | | |
| No | 162 (24.7) | 4 (2.5) | 158 (97.5) | 0.368 |
| Yes | 493 (75.3) | 21 (4.3) | 472 (95.7) | |
| **MNP supplementation** | | | | |
| No | 598 (90.6) | 20 (3.3) | 578 (96.7) | 0.124 |
| Yes | 62 (9.4) | 5 (8.1) | 57 (91.9) | |
| **Deworming** | | | | |
| No | 490 (76.0) | 17 (3.5) | 473 (96.5) | 0.858 |
| Yes | 155 (24.0) | 7 (4.5) | 148 (95.5) | |
| **Diarrhea** | | | | |
| No | 503 (75.4) | 17 (3.4) | 486 (96.6) | 0.496 |
| Yes | 164 (24.6) | 8 (4.9) | 156 (95.1) | |
| **Fever** | | | | |
| No | 439 (65.8) | 17 (3.9) | 422 (96.1) | 0.842 |
| Yes | 228 (34.2) | 8 (3.5) | 220 (96.5) | |
| **Respiratory tract infection** | | | | |
| No | 430 (64.5) | 20 (4.7) | 410 (95.3) | 0.167 |
| Yes | 237(35.5) | 5 (2.1) | 232 (97.9) | |
| **Anaemia Hb level <11 g / dL** | | | | |
| No | 307 (71.9) | 16 (5.2) | 291 (94.8) | 0.461 |
| Yes | 120 (28.1) | 4 (3.3) | 116 (96.7) | |
| **Maternal Se status** | | | | |
| Deficient | 479 (70.1) | 6 (1.3) | 473 (98.7) | <0.001* |
| Adequate | 204 (29.9) | 20 (9.8) | 184 (90.2) | |

**Notes**: HAZ, height-for-age Z-score; WHZ, weight-for-height Z-score; WAZ, weight-for-age Z-score; SD, Standard deviation;

§ Plasma Se concentration level < 70 μg/L;

‡Plasma Se concentration ≥70 μg/L;

*Significance level P <0·05;

[†]P value from Pearson's Chi square test.

**Table 4. Nutritional status of WRA in rural Zimbabwe by Se status.**

| Variable | Total n (% of category) | ‡Se–adequate n (%) | §Se deficient n (%) | P-value† |
|---|---|---|---|---|
| **Anaemia Status (haemoglobin g/dL)** | | | | |
| <12 | 88 (19.8) | 30 (34.1) | 58 (65.9) | 0.463 |
| ≥12 | 357 (80.2) | 138 (38.7) | 219 (61.3) | |
| **Height (cm)** | | | | |
| <145 | 8 (1.2) | 4 (50.0) | 4 (50.0) | 0.248 |
| ≥145 | 675 (98.8) | 200 (29.6) | 475 (70.4) | |
| **Body Mass Index (kg/m2)** | | | | |
| <18.5 | 74 (10.8) | 16 (21.6) | 58 (78.4) | 0.108 |
| ≥18.5 | 609 (89.2) | 188 (30.9) | 421 (69.1) | |
| **Nutritional Status** | | | | |
| **Underweight** | 74 (10.9) | 16 (8.0) | 58 (78.4) | |
| **Normal weight** | 353 (52.1) | 100 (28.3) | 253 (71.7) | |
| **Overweight** | 180 (26.6) | 59 (32.8) | 121(67.2) | 0.087 |
| **Class I obese** | 48 (7.1) | 21 (43.8) | 27 (56.2) | |
| **Class II obese** | 13 (1.9) | 2 (15.4) | 11 (84.6) | |
| **Morbid obese** | 9 (1.3) | 2 (22.2) | 7 (77.8) | |
| **Prevalence of Se deficiency** | 683 (100) | 204 (29.9) | 479 (70.1) | |

Notes:

§ Plasma Se concentration <70 μg / L;

‡ Plasma Se concentration ≥70 μg / L;

†P value significant at p < 0.05 from Pearson's Chi-square test.

household food allocation for children, which is affected by cultural norms in society and women's empowerment in households [67]. In Zimbabwe, sex vulnerability to MNDs is not considered in micronutrient supplementation programming for children aged 6–59 months. Currently, there is a provision of multiple micronutrient powders (MNPs) for point-of-use fortification, a blanket program targeted for children 6–23 months in select districts. Sex variabilities could be implemented in MNP supplementation, as seen in growth monitoring, where growth charts target individual sexes [44]. Multiple micronutrient powders targeted for girls would contain higher Se concentrations >17 μg/g [68] relative to boys, within the upper tolerable limit for Se of 400 μg/day [69]. The recommendation, therefore, would be to conduct further studies at scale to validate the sex disparity and potentially increase the Se concentration of MNPs targeted for girls.

## 4.3. Maternal Se status and risk of deficiency in children

In this study, maternal Se deficiency was positively associated with childhood Se deficiency. The prevalence of Se deficiency in children from Se deficient WRA was four times higher compared to their counterparts. Intergenerational transmission of micronutrient status was observed in Malawi, Mozambique, Namibia [70] and Zimbabwe [37]. Selenium plays a significant role in female reproductive processes [71] and its deficiency during pregnancy and lactation influences nutrition outcomes in children [72–74]. Our results indicated a lower prevalence of Se deficiency in younger children and a higher prevalence in older children among Se-deficient mothers this can be attributed to the protective effect of breastfeeding. The current study shows a high prevalence of breastfeeding, with an exclusive breastfeeding rate

**Table 5. Predictors of Se deficiency among children aged 6–59 months and WRA from rural Zimbabwe.**

| Variable | B | S.E. | P value[†] | OR | 95% C.I. | |
|---|---|---|---|---|---|---|
| | | | | | Lower | Upper |
| **Children (6–59 months)** | | | | | | |
| **Murewa district** No = 0; Yes = 1 | 0.29 | 0.32 | 0.37 | 1.33 | 0.71 | 2.50 |
| **Se deficient mother** Adequate = 0; Deficient = 1 | 1.45 | 0.52 | 0.005* | 4.25 | 1.55 | 11.67 |
| **Being a girl** (Male = 0; Female = 1) | 1.05 | 0.50 | 0.035* | 2.84 | 1.08 | 7.51 |
| **Unimproved dug well as a water source** (Improved dug well = 0; unimproved = 1) | -0.07 | 0.91 | 0.939 | 0.93 | 0.16 | 5.59 |
| **Maize production for consumption** (No = 0; Yes = 1) | -0.84 | 0.84 | 0.319 | 0.43 | 0.08 | 2.25 |
| **Cowpea production for consumption** (No = 0; Yes = 1) | -0.79 | 1.18 | 0.515 | 0.46 | 0.05 | 4.59 |
| **Groundnut production for consumption** (No = 0; Yes = 1) | 0.184 | 1.037 | 0.859 | 1.20 | 0.16 | 9.18 |
| **Sugar bean production for consumption** (No = 0; Yes = 1) | -0.46 | 1.09 | 0.675 | 0.63 | 0.08 | 5.36 |
| **Sweet potato production for consumption** (No = 0; Yes = 1) | -0.72 | 1.24 | 0.560 | 0.49 | 0.04 | 5.50 |
| **Onion production for consumption** (No = 0; Yes = 1) | -0.98 | 1.23 | 0.422 | 0.37 | 0.03 | 4.13 |
| **Women of Reproductive Age** | | | | | | |
| **Murewa District** No = 0; Yes = 1 | -0.15 | 0.13 | 0.233 | 0.86 | 0.68 | 1.10 |
| **Unimproved dug well as a water source** (improved = 0; unimproved = 1) | -0.05 | 0.26 | 0.850 | 0.95 | 0.57 | 1.59 |
| **Reproductive age 18–34 years** (<18≥35 years = 0; 18–34 years = 1) | 0.02 | 0.19 | 0.927 | 1.02 | 0.71 | 1.47 |
| **Monthly income below 10 USD** (≥10 USD = 0; <10USD = 1) | -0.55 | 0.44 | 0.213 | 0.58 | 0.25 | 1.37 |
| **Maize production for consumption** (No = 0; Yes = 1) | -0.75 | 0.329 | 0.022* | 0.47 | 0.247 | 0.897 |
| **Cowpea production for consumption** (No = 0; Yes = 1) | -0.14 | 0.35 | 0.697 | 0.87 | 0.44 | 1.73 |
| **Groundnut production for consumption** (No = 0; Yes = 1) | -0.11 | 0.35 | 0.755 | 0.90 | 0.45 | 1.78 |
| **Sugar bean production for consumption** (No = 0; Yes = 1) | -0.26 | 0.31 | 0.396 | 0.77 | 0.43 | 1.40 |
| **Sweet potato production for consumption** (No = 0; Yes = 1) | -0.32 | 0.34 | 0.335 | 0.72 | 0.38 | 1.40 |
| **Onion production for consumption** (No = 0; Yes = 1) | 0.00 | 0.3 | 0.999 | 1.00 | 0.56 | 1.80 |

**Notes**: Selenium deficient mother; Plasma Se concentration <70 μg / L; Selenium adequate mother; Plasma Se concentration ≥70 μg / L,

[†]P value from multiple logistic regression analysis;

*Significant at P<0·05, by the Enter regression model.

higher than the WHO target (50%) and the global average (44%) [75]. Most Zimbabwean mothers breastfeed their babies for up to 2 years and sometimes beyond [32]. Breastmilk is an important source of Se providing median Se concentrations of up to 15–26 μg/L [76]. The Se concentration and glutathione peroxidase activity in human milk is influenced directly by the

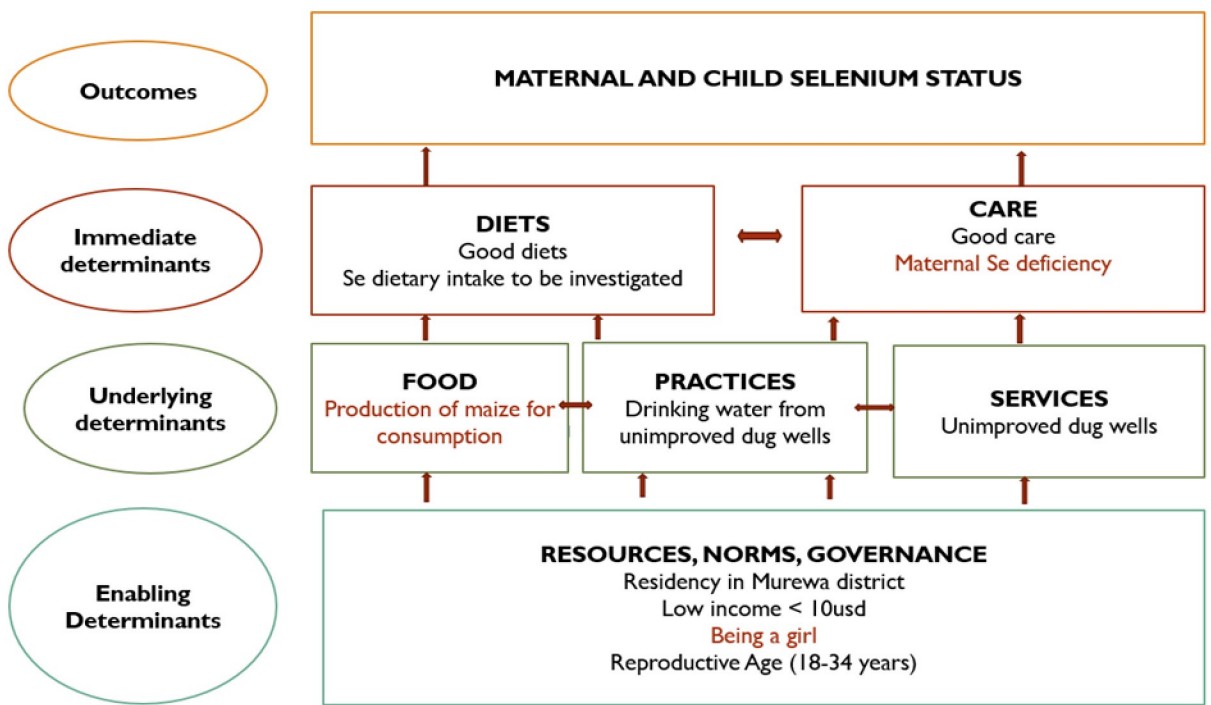

**Fig 2. Maternal and child plasma Se deficiency causal framework predictors (red) and determinants (black) for Murewa, Shamva, and Mutasa districts, rural Zimbabwe.**

Se intake of the mother [76–79]. Furthermore, the current study also showed that WRA 18–34 years were the most Se deficient, attributable to the depletion of micronutrient stores due to successive pregnancies and breastfeeding episodes common at this optimal reproductive phase [73]. Selenium concentrations were reported to be low in women of reproductive age in the United Kingdom, decreasing further during pregnancy, thus resulting in low plasma and placental antioxidant enzyme activities [80]. Multiple micronutrient supplementation during pregnancy indicated an upward trend in Se concentration across semesters [81]. The recommendation for Zimbabwe is to scale up the MNP supplementation for children 6–59 months and to initiate maternal multiple micronutrient supplementation for WRA in addition to iron and folate supplementation currently being given during pregnancy, from preconception through lactation, proven to reduce MNDs in both children and women [82]. Further research into placental transfer of Se in utero, diet quality, and birth order which can affect Se sufficiency/insufficiency postpartum is recommended to validate the findings of the current study.

## 4.4. Maize production for consumption increases the risk of Se deficiency

Primarily, the production of maize for consumption predicted low Se status in women. Maize is the staple crop in Zimbabwe with an estimated maize production of over two million metric tonnes [83]. The average maize consumption for adults in Zimbabwe is over 250 g/person/day, with an energy supply accounting for ~3500 kJ/capita/day [84, 85]. The high consumption of maize makes it a major contributor to dietary Se [29, 86], consumption of 100 grams/person/day of maize contributes to MNDs, Se included [85]. Similar findings were reported in Malawi where Se deficiency is widespread [22], potentially as a result of a Se-deficient maize crop [86], and might consequently be mirrored with human Se status [87]. In Ethiopia, the risk of

human Se deficiency was also associated with the staple diet [88]. Based on our findings, Se agronomic biofortification of the staple crop with fertiliser or point-of-use fortification may be necessary. Minute quantities of Se are required to result in meaningful contributions to grain Se concentration and dietary intake. Studies in Malawi and Ethiopia have shown that agronomic biofortification of staple crops with 20 g/hectare Se has the potential to increase grain Se concentration [89, 90]. Cognisant of that, soil Se concentrations are the primary driver of population Se status, future research should involve the examination of soil, crop, and dietary Se concentrations (for example whole foods using total diet data) to define whether there is a generalized deficiency in these regions. This may be particularly relevant given homestead farming may not regulate the use of fertilizers and/or biofortified crops. Further to this, maize grain in Zimbabwe has been implicated with mycotoxin contamination mainly *fusarium* [91]. *Fusarium* produces T-2 toxin as a secondary metabolite whose synergistic effects with Se deficiency pose potential detrimental health hazards to the Se-deficient population [92]. Investigation of this correlation might be warranted.

## 5. Limitations of study

The current study is exploratory, tests of statistical significance should be interpreted with caution as false positives may occur due to multiple comparisons. Further studies are warranted to confirm the results. Additionally, the study did not include dietary assessments that could have provided evidence on the foods contributing to Se intake, to assess implications for human Se status. Regardless, the present study contributes to our knowledge of the association between maternal and child Se deficiency.

## 6. Conclusions

The current study showed that being a girl and maternal Se deficiency were positively associated with Se deficiency in children aged 6–59 months while maize crop production was positively associated with Se deficiency in WRA. Interventions that focus on improving conceptual and maternal nutritional status, micronutrient supplementation, and biofortification may be important strategies to reduce Se deficiency in vulnerable populations from low and lower-middle-income countries in Africa.

## Supporting information

**S1 Table. Correlation between plasma Se concentration and acute phase proteins.** (DOCX)

**S2 Table. Correlation between child and maternal Se status stratified by child age group.** (DOCX)

**S1 Data.** (ZIP)

## Acknowledgments

The authors thank all the parents and caregivers of the infants for participating in the study, and the entire Zimbabwe GeoNutrition field team for executing the study. Gratitude to the Ministry of Health and Child Care (MoHCC) authorities for their collaboration and support; ZIMSTAT for the household listing and mapping exercise, BMGF GeoNutrition for staff time and Martin R. Broadley for co-designing the study methodology.

## Author Contributions

**Conceptualization:** Beaula Mutonhodza, Elizabeth H. Bailey, Tonderayi M. Matsungo.

**Data curation:** Beaula Mutonhodza, Tonderayi M. Matsungo, Prosper Chopera.

**Formal analysis:** Beaula Mutonhodza, Elizabeth H. Bailey, Tonderayi M. Matsungo, Prosper Chopera.

**Funding acquisition:** Elizabeth H. Bailey.

**Investigation:** Beaula Mutonhodza, Muneta G. Manzeke-Kangara, Elizabeth H. Bailey, Tonderayi M. Matsungo, Prosper Chopera.

**Methodology:** Beaula Mutonhodza, Muneta G. Manzeke-Kangara, Elizabeth H. Bailey, Tonderayi M. Matsungo, Prosper Chopera.

**Project administration:** Beaula Mutonhodza, Muneta G. Manzeke-Kangara, Tonderayi M. Matsungo, Prosper Chopera.

**Resources:** Muneta G. Manzeke-Kangara, Elizabeth H. Bailey.

**Supervision:** Elizabeth H. Bailey, Tonderayi M. Matsungo, Prosper Chopera.

**Validation:** Muneta G. Manzeke-Kangara, Elizabeth H. Bailey, Tonderayi M. Matsungo, Prosper Chopera.

**Visualization:** Beaula Mutonhodza, Tonderayi M. Matsungo, Prosper Chopera.

**Writing – original draft:** Beaula Mutonhodza.

**Writing – review & editing:** Muneta G. Manzeke-Kangara, Elizabeth H. Bailey, Tonderayi M. Matsungo, Prosper Chopera.

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
