## [Decision Letter · Decision Letter 0]

26 Apr 2024

PGPH-D-24-00427

Maternal selenium deficiency was positively associated with the risk of selenium deficiency in children aged 6-59 months in rural Zimbabwe

Dear Dr. Mutonhodza,

Thank you for submitting your manuscript to PLOS Global Public Health. Firstly, we would like to apologize for the delay in processing your manuscript. It has been exceptionally difficult to secure reviewers to evaluate your study. We have now received one completed review, which is available below. The reviewer has raised significant scientific concerns about the study that need to be addressed in a revision.

Please note that we have only been able to secure a single reviewer to assess your manuscript. We are issuing a decision on your manuscript at this point to prevent further delays in the evaluation of your manuscript. Please be aware that the editor who handles your revised manuscript might find it necessary to invite additional reviewers to assess this work once the revised manuscript is submitted. However, we will aim to proceed on the basis of this single review if possible.

We look forward to receiving your revised manuscript.

Kind regards,

Miquel Vall-llosera Camps, Ph.D.

Staff Editor

Journal Requirements:

Reviewers' comments:

Reviewer's Responses to Questions

**Comments to the Author**

1. Does this manuscript meet PLOS Global Public Health’s publication criteria? Is the manuscript technically sound, and do the data support the conclusions? The manuscript must describe methodologically and ethically rigorous research with conclusions that are appropriately drawn based on the data presented.

Reviewer #1: Yes

2. Has the statistical analysis been performed appropriately and rigorously?

Reviewer #1: Yes

3. Have the authors made all data underlying the findings in their manuscript fully available (please refer to the Data Availability Statement at the start of the manuscript PDF file)?

Reviewer #1: Yes

4. Is the manuscript presented in an intelligible fashion and written in standard English?

Reviewer #1: Yes

5. Review Comments to the Author

Reviewer #1: Thank you for the opportunity to review the paper titled: “Maternal selenium deficiency was positively associated with the risk of selenium deficiency in children aged 6-59 months in rural Zimbabwe”.

The paper is well written and shares some interesting information about selenium concentrations in human specimens and its correlates.

Recommend major revisions.

Line 49: Can you explain why? How and why does HIV exacerbate Se deficiency in SSA?

Lines 134-135: For inflammation adjustment and Se, can you kindly explain why you have specifically noted negative correlations Is this expected based on the literature, if so, please cite biological plausibility and theory to explain why? Would you not apply correction if the relationships were positive? Kindly define the negative acute phase response in a little more detail with relevant citations.

Lines 1222-135: Can you please define what % of your blood samples were below the LOD/LOQ? How did you handle thresholded data? Was any imputation used to account for values < LOD (i.e., low (0), medium (1/2 LOD), high (LOD) imputation? If no imputation was used, kindly define what % of your samples were below the LOD/LOQ at the least and how you handled these observations?

Line 146: Why did you use a p < 0.10 cut-off to define significance? Can you kindly also look at your data using linear regression i.e., continuous plasma Se to define how your covariates respond when using the variable without dichotomizing it. Can you also show your correlation analyses in some form of a correlation matrix so the reader can see what your coefficients looked like, before proceeding to logistic regression.

Table 1: These data seem to be clearly pointing to soil insufficiency for Se. Given Se is primarily found in soil and insufficiency in concentrations in the soil will lead to population deficiency, it is very clear that there should be more work done to examine crop concentrations of Se to define whether there is generalized insufficiency/deficiency in these regions. This may be particularly relevant given homestead farming may not regulate use of fertilizers and/or biofortified crops. You have commented on this in your discussion, but you may want to discuss limitations in more detail later on in your text and/or suggest how you would extend these analyses in terms of next steps which could involve examining soil, crop and dietary Se concentrations (whole foods using total diets data for e.g. since deficiency is so high in the population).

Table 1/Lines 185-186: There may be a need to discuss in the introduction, what foods in the typical Zimbabwean diet are rich in Se and why you are seeing such strong significant associations between production of these staples and Se deficiency – are these food typically low in Se (globally and in Zimbabwe and if so why?).

Lines 191-192: Can you look at dietary data to examine how they correlate to Se deficiency and Se concentrations using linear and logistic regression.

Lines 200-202: This sentence needs to be clarified. Can you please include the accompanying Se deficiency % in mother-child dyads?

Table 4: For the overweight/obese cells before line 213, wouldn’t the 0.087 p-value be significant as per the criteria you have described in your methods.

Table 5: Please define your reference values for table 5

Lines 228-238: The issue of sex specific differences needs to be explained further.

Selenium Metabolism, Regulation, and Sex Differences in Mammals | SpringerLink

Selenium is concentrated in male gonads which could possibly explain why male children had lower deficiency had than female children – given what is known of selenium saturation in the human body. Kindly refer to the text above for more. Could you attribute your findings to sex differences in infant and young child feeding practices/breastfeeding behaviours and gendered preferences for male children, if culturally relevant? Can you comment on diet quality and differences between male vs. female children if relevant?

Line 252-253: Sentence is incomplete.

Line 257-261: You may consider looking into how placental transfer of Se in utero can affect selenium sufficiency/insufficiency postpartum. Since you see clear statistically significant trends in maternal and child Se deficiency, you may want to examine diet quality which is a likely confounder, given it would affect Se status in mothers and children. Since you have younger children in this study (6-12 months), you may consider stratifying children by age group to see if the younger children have stronger correlation with maternal Se status compared to the older children. Additionally, you may consider discussing birth order of the children involved.

Line 257-261: What is known of Se concentrations in the 6-12 month age group, given you are not discussing a supplementation study here? We know Se concentrations decrease during pregnancy and at delivery and then level off, can you comment on this matter and how it could be impacting the strong correlations you see in mother-infant dyads if relevant?

Line 285-287: You may consider examining whether mycotoxin contamination in crops in Zimbabwe is responsible for low Se concentrations. T-2 toxin co-occurs with Se deficiency in soil and could be an explanatory factor if relevant in your context, in addition to other mycotoxins present in crops including maize and groundnuts. There may be other mycotoxins of relevance in addition to T-2 toxin. Kindly briefly examine the literature to see if this is a relevant explanatory factor. You may consider examining the work of Laura Smith and Francis Ngure (Zwitambo) for more.

6. PLOS authors have the option to publish the peer review history of their article (what does this mean?). If published, this will include your full peer review and any attached files.

**Do you want your identity to be public for this peer review?** For information about this choice, including consent withdrawal, please see our Privacy Policy.

Reviewer #1: No

---

## [Decision Letter · Decision Letter 1]

29 May 2024

Maternal selenium deficiency was positively associated with the risk of selenium deficiency in children aged 6-59 months in rural Zimbabwe

PGPH-D-24-00427R1

Dear Mrs Mutonhodza,

We are pleased to inform you that your manuscript 'Maternal selenium deficiency was positively associated with the risk of selenium deficiency in children aged 6-59 months in rural Zimbabwe' has been provisionally accepted for publication in PLOS Global Public Health.

Best regards,

Shaonong Dang, PhD

Academic Editor

Authors have addressed most of the comments, and the manuscript has been improved much for publication.

Reviewer Comments (if any, and for reference):

Reviewer's Responses to Questions

**Comments to the Author**

1. If the authors have adequately addressed your comments raised in a previous round of review and you feel that this manuscript is now acceptable for publication, you may indicate that here to bypass the “Comments to the Author” section, enter your conflict of interest statement in the “Confidential to Editor” section, and submit your "Accept" recommendation.

Reviewer #1: All comments have been addressed

2. Does this manuscript meet PLOS Global Public Health’s publication criteria? Is the manuscript technically sound, and do the data support the conclusions? The manuscript must describe methodologically and ethically rigorous research with conclusions that are appropriately drawn based on the data presented.

Reviewer #1: Yes

3. Has the statistical analysis been performed appropriately and rigorously?

Reviewer #1: Yes

4. Have the authors made all data underlying the findings in their manuscript fully available (please refer to the Data Availability Statement at the start of the manuscript PDF file)?

Reviewer #1: Yes

5. Is the manuscript presented in an intelligible fashion and written in standard English?

Reviewer #1: Yes

6. Review Comments to the Author

Reviewer #1: (No Response)

7. PLOS authors have the option to publish the peer review history of their article (what does this mean?). If published, this will include your full peer review and any attached files.

**Do you want your identity to be public for this peer review?** For information about this choice, including consent withdrawal, please see our Privacy Policy.

Reviewer #1: No
